# Behavioural correlations of the domestication syndrome are decoupled in modern dog breeds

Christina Hansen Wheat[1], John L. Fitzpatrick[1], Björn Rogell[1] & Hans Temrin[1]

Domestication is hypothesized to drive correlated responses in animal morphology, physiology and behaviour, a phenomenon known as the domestication syndrome. However, we currently lack quantitative confirmation that suites of behaviours are correlated during domestication. Here we evaluate the strength and direction of behavioural correlations among key prosocial (sociability, playfulness) and reactive (fearfulness, aggression) behaviours implicated in the domestication syndrome in 76,158 dogs representing 78 registered breeds. Consistent with the domestication syndrome hypothesis, behavioural correlations within prosocial and reactive categories demonstrated the expected direction-specificity across dogs. However, correlational strength varied between dog breeds representing early (ancient) and late (modern) stages of domestication, with ancient breeds exhibiting exaggerated correlations compared to modern breeds across prosocial and reactive behaviours. Our results suggest that suites of correlated behaviours have been temporally decoupled during dog domestication and that recent shifts in selection pressures in modern dog breeds affect the expression of domestication-related behaviours independently.

---

[1] Department of Zoology, Stockholm University, SE-10691 Stockholm, Sweden. Correspondence and requests for materials should be addressed to C.H.W. (email: christina.hansen@zoologi.su.se)

The domestication of plants and animals provides an ideal framework for studying evolutionary responses to selection[1]. Domesticated species are selected to live in environments shaped and controlled by humans[2–4], often affecting the same traits across a wide range of species[5–8]. Domesticated species typically display a 'domestication syndrome', exhibiting similar patterns of simultaneous alterations in physiology and morphology, and for animals, behaviour compared with their wild counterparts[5,9]. For example, domestication in mammalian species is commonly associated with reductions in brain size, depigmentation, increased tameness and changes in hormone and neurotransmitter levels[1,10]. While the mechanistic basis of the domestication syndrome remains a source of controversy[8,10], the notion of correlated changes in traits is now embedded as a standard paradigm for studying domestication[7,11]. Correlations among traits may imply a common mechanism underlying trait expression[8], or alternatively result from sequential, and not simultaneous, trait changes during domestication[12]. Hence, disentangling the observation of correlated traits in contemporary domesticated species from the selective processes that drove divergence from wild progenitors remains a major challenge when studying domestication.

Though behavioural selection is likely at the forefront of the domestication process[7,13,14], understanding how domestication influences behaviours in animals represents a particular challenge, since archaeological records of behaviours do not exist. In a classic selection experiment on foxes (*Vulpes vulpes*), Belyaev and colleagues[7,13] demonstrated that selection on tame behaviour alone was sufficient to drive correlated physiological and morphological changes characteristic of the domestication syndrome[15]. More generally, domesticated animals commonly express increased levels of prosocial behaviours (sociability and playfulness), alongside lowered levels of reactive behaviours (fear and aggression), when compared with their wild ancestors[11,14,16,17]. Simultaneous alterations of behavioural traits can arise if behaviours are controlled by the same physiological mechanisms[8,18] or be based on genetic control of the traits[19,20] (i.e. pleiotropy or linkage disequilibrium[21,22]). In a range of domesticated mammals (e.g. foxes[7,23], rats[24], guinea pigs[16]), the shift from reactive to prosocial phenotypes is associated with lower reactivity of the hypothalamic–pituitary–adrenocortical (HPA) axis, which is related to adrenalin and stress responses[7,8,23,25,26], and neural crest cell deficits during embryonic development[8]. Behavioural changes observed in domesticated animals have therefore been hypothesized to originate from a shared physiological mechanism, leading to the general assumption that behavioural traits change in a correlated fashion during domestication[7,14,15,26]. Moreover, on the level where selection is applied, correlated selection may be an important force shaping within-group level correlations by driving genetic correlations among traits[27]. Given these mechanisms, the correlated behavioural changes observed during domestication are analogous to more commonly studied behavioural syndromes[19]. Yet, behavioural alterations brought about by domestication have predominantly been studied separately or pairwise[16,17,28], rather than in concert as a syndrome[19,20]. Consequently, it remains unclear whether the simultaneous alterations of the prosocial and reactive behavioural traits associated with the domestication syndrome are consistently expressed throughout the course of domestication.

The domestic dog (*Canis familiaris*) is an excellent model species to investigate questions concerning domestication. Originating from the grey wolf (*Canis lupus*) at least 15,000 years ago, the dog was the first animal to be domesticated[1,29,30]. The established divergence between dog breeds and their wild ancestor presents an ideal opportunity for temporal comparison[31,32], as dog breeds can be divided into two major groups, ancient and modern breeds[31]. Ancient breeds are a small group of dog breeds originating more than 500 years ago, characterized by detectable genetic admixture with wolves and represent an early stage of dog domestication[31]. Modern breeds, which represent the vast majority of the more than 400 present day dog breeds, originated from stringent breeding efforts taking place only over the last 200 years[31,33], creating strong, breed-specific selection pressures on behavioural and morphological traits[34–38]. These modern breeds are highly divergent from both the ancient breeds and wolves, with no detectable wolf admixture[31]. Comparisons of correlated behavioural changes between ancient and modern breeds can therefore offer insights into correlated behavioural responses to alternative selective pressures and/or the temporal underpinnings of behavioural evolution during dog domestication.

Here we use a standardized behavioural test applied to over 76,000 dogs, representing 78 registered dog breeds, to test the hypothesis that behaviours implicated in the domestication syndrome are correlated and consistently expressed throughout the domestication of dogs. Since domesticated animals generally exhibit a concurrent increase in prosocial and reduction in reactive behaviours, there is a general assumption that behaviours change in a correlated fashion during domestication[7,11,14,16,17]. Specifically, in a long-running selection experiment targeting tame behaviour, selected foxes exhibited increased sociability towards humans (e.g. tail wagging) and play behaviour, alongside reduced aggression and fear[7,13,39]. Further, work on foxes[7,13] and other domesticated animals (e.g. rats[24], guinea pigs[16]) has demonstrated that selection for tame and/or against aggressive behaviours leads to reduced stress responses and correlated responses in the physiological parameters controlling neurotransmitters associated with reductions in fear and aggression[40]. Together, these observations across domesticated animals have led to the general prediction that behaviours within prosocial and reactive categories will be positively co-expressed, while behaviours across prosocial and reactive categories will be negatively co-expressed[7,13,14,16,40,41]. In addition, behavioural correlations are predicted to be stable if they are expressed through a shared physiological or genetic mechanism[19,42,43]. Thus, if behavioural correlations in the domestication syndrome are generated via a shared underlying mechanism (e.g. attenuation of the hypothalamic–pituitary HPA axis or neural crest deficit), we would, according to the domestication syndrome hypothesis, expect a general pattern of positive correlations within each behavioural category (prosocial and reactive) and negative correlations across behavioural categories (prosocial vs. reactive). However, behavioural correlations can respond adaptively to altered selection pressures[19] and breakdown under changed environmental conditions or selection regimes[19,44–47]. Therefore, varying selection pressures at different times during dog domestication could cause a decoupling (i.e. destabilization) of the behaviours in the domestication syndrome[15]. If so, we would expect to find discrepancy in the behavioural correlations between ancient and modern breeds. Here, we test the direction-specific hypothesis among behavioural categories arising from the domestication syndrome by directly evaluating the prediction that prosocial and reactive behaviours will exhibit positive covariance within behavioural categories, but negative covariance between behavioural categories. We then evaluate the temporal stability of behavioural correlations during dog domestication. Our results broadly support the predictions of the domestications syndrome hypothesis, but suggest that the extensive, recent breeding efforts applied to modern breeds has decoupled the covariance between prosocial and reactive behaviours in modern dog breeds.

## Results

**Behavioural correlations in ancient and modern dog breeds.** We analysed data from the Swedish Kennel Club's (a national organization dedicated to breeding, showing and training of dogs in Sweden) database on 76,158 dogs representing 78 registered dog breeds (7 ancient and 71 modern breeds; Fig. 1a) tested using the Dog Mentality Assessment (DMA) between 1997 and 2013. The DMA is widely used on dogs in Sweden to quantify and assess a range of behaviours[48,49] and to give dog owners and breeders information on behaviours relevant for breeding and training. In the DMA, dogs are exposed to 10 different standardized situations, in which various behavioural reactions are assessed on a standardized scale. For this study we selected key prosocial (sociability and playfulness) and reactive (fearfulness and aggression) behavioural traits from the DMA that are relevant to the domestication syndrome hypothesis: (1) sociability, in which the dog is greeted and examined by a stranger, (2) playfulness, in which the dog is invited, by the stranger, to play tug of war with a rag, (3) fearfulness, which is measured in the two different contexts; sudden threat (a startle stimulus is presented suddenly to the dog at a close distance) and persistent threat (two people dressed as ghosts slowly approach the dog from a distance of 20 metres), and (4) aggression, which is measured in the three different contexts; distant threat (a person dressed in a cape moves erratically and claps their hands 40 metres from the dog, eventually approaching the dog), sudden threat (as described in Fearfulness—sudden threat) and persistent threat (as described in Fearfulness—persistent threat). See Supplementary Table 1 for further description of the behavioural assays.

Using these key behaviours quantified in the DMA, we set out to test for behavioural correlations consistent with the domestication syndrome in dogs. The domestication syndrome hypothesis makes clear a priori predictions regarding the direction of behavioural correlations (both predicted positive and negative correlations between prosocial and reactive behaviours, see Introduction). To achieve our main aim of assessing direction-specific differences in behavioural correlations between ancient and modern breeds, we first extracted residual values from behavioural test scores for all dogs, while accounting for potentially confounding variables that arise from the expansive nature of our data set (e.g. effects of sex, age, geographical location and identity of the observer scoring the behaviours). For each breed we then generated a correlation matrix based on these residual behavioural scores (these values represent the within-breed, among-individual correlations calculated from residual values), with correlation coefficients ($r$) for each pairwise combination of behaviours. Using these correlations coefficients, and their associated sample sizes, we calculated $Z_r$ values using Fisher's transformation weighted by sample size and the sampling variances ($V_z$) for each behavioural correlation and breed (Supplementary Data 1, 2 and 3). This approach allowed us to generate sample size-corrected effect sizes that indicate the strength and direction of associations between behaviours. We then tested if within-breed correlations predicted to be either positive or negative went in the direction predicted by the domestication syndrome hypothesis and if the strength of behavioural correlations differed between ancient and modern breeds by fitting the $Z_r$ and $V_z$ values in a phylogenetically-corrected meta-analytical model that included breed and the type of behavioural correlation as random effects.

The phylogenetically corrected meta-analyses revealed that the expected positive within-breed behavioural correlations were significantly different from zero in the direction consistent with our a priori expectation based on the theoretical framework of the domestication syndrome hypothesis (Table 1, Fig. 1b). These results indicate that correlations within prosocial behaviours and

reactive behaviours exhibit overall positive effect sizes in dogs. We also detected a significant interaction between breed type (ancient vs. modern) and the expected direction of the behavioural correlation (positive vs. negative), indicating that ancient breeds had more exaggerated effect sizes, but this exaggeration was limited to expected negative behavioural correlations between prosocial and reactive behaviours (Table 1, Fig. 1b). Among modern dog breeds, within-breed correlations between prosocial and reactive behaviours did not differ significantly from zero (Fig. 1b). These findings suggest that the strength of the expected within-breed negative correlations between prosocial and reactive behaviours weakened over the domestication history of dogs. Thus, exaggeration of behavioural effect sizes in ancient breeds is dependent on the type of behavioural correlations examined and the a priori predictions from the domestication syndrome hypothesis.

While the domestication syndrome hypothesis makes generic predictions about the direction of behavioural correlations, whether the pattern detected above demonstrates a general pattern in the correlational structure of the behavioural data, as opposed to being driven by behaviour-specific correlations, remains unclear. To explore behaviour-specific correlations we visualized the difference in $Z_r$ values between ancient and modern breeds for each pairwise combination of behaviours in meta-analytical models. Among expected negative correlations, we found five correlations that were significantly different from zero in the expected negative direction (i.e. 95% CIs of the model intercept not overlapping zero, Supplementary Figure 1, Supplementary Table 4), with effect sizes typically exaggerated in ancient compared with modern breeds. Thus, the exaggerated effect sizes detected in ancient breeds in the direction-specific analyses appear to arise from a general exaggeration in the effect size magnitude of behavioural correlations for ancient breeds across the behaviours considered. Among the expected positive correlations, only two correlations significantly differed from zero, both in the expected direction (Supplementary Figure 1, Supplementary Table 4). These two positive correlations had among the strongest effect sizes in our study, and likely drove the overall positive effect sizes we detected in the positive direction-specific analysis above (Supplementary Figure 1, Supplementary Table 4).

The analyses above focused on the strength of behavioural correlations within-breeds, as this is where selection is applied during the domestication process with the intention of changing behaviour (i.e. breed mean values). Therefore, reductions in within-breed behavioural correlations should consequently influence correlations in mean behavioural values among breeds. To assess this possibility, we compared the strength of the a priori direction of mean correlation values between behaviours for ancient and modern breeds (Supplementary Table 5). Among ancient breeds, 15 of the 17 behavioural correlations based on breed mean values were in the direction predicted by the domestication syndrome hypothesis, a value that differs from chance (Exact binomial test, $p = 0.002$, Supplementary Table 5). In contrast, in modern dog breeds roughly half (9 of the 17) of the behavioural correlations based on breed mean values did not match the expected direction from the domestication syndrome (Exact binomial test, $p = 1.0$, Supplementary Table 5). When we compared the strength of the behavioural correlations based on breed mean values between ancient and modern breeds, we found that ancient breeds had exaggerated direction-specific correlation values significantly more frequently than did modern breeds (14 of the 17 correlations were exaggerate in a direction-specific manner in ancient breeds; Exact binomial test, $p = 0.01$, Supplementary Table 5). These analyses of behavioural correlations based on breed mean values reinforce our main findings

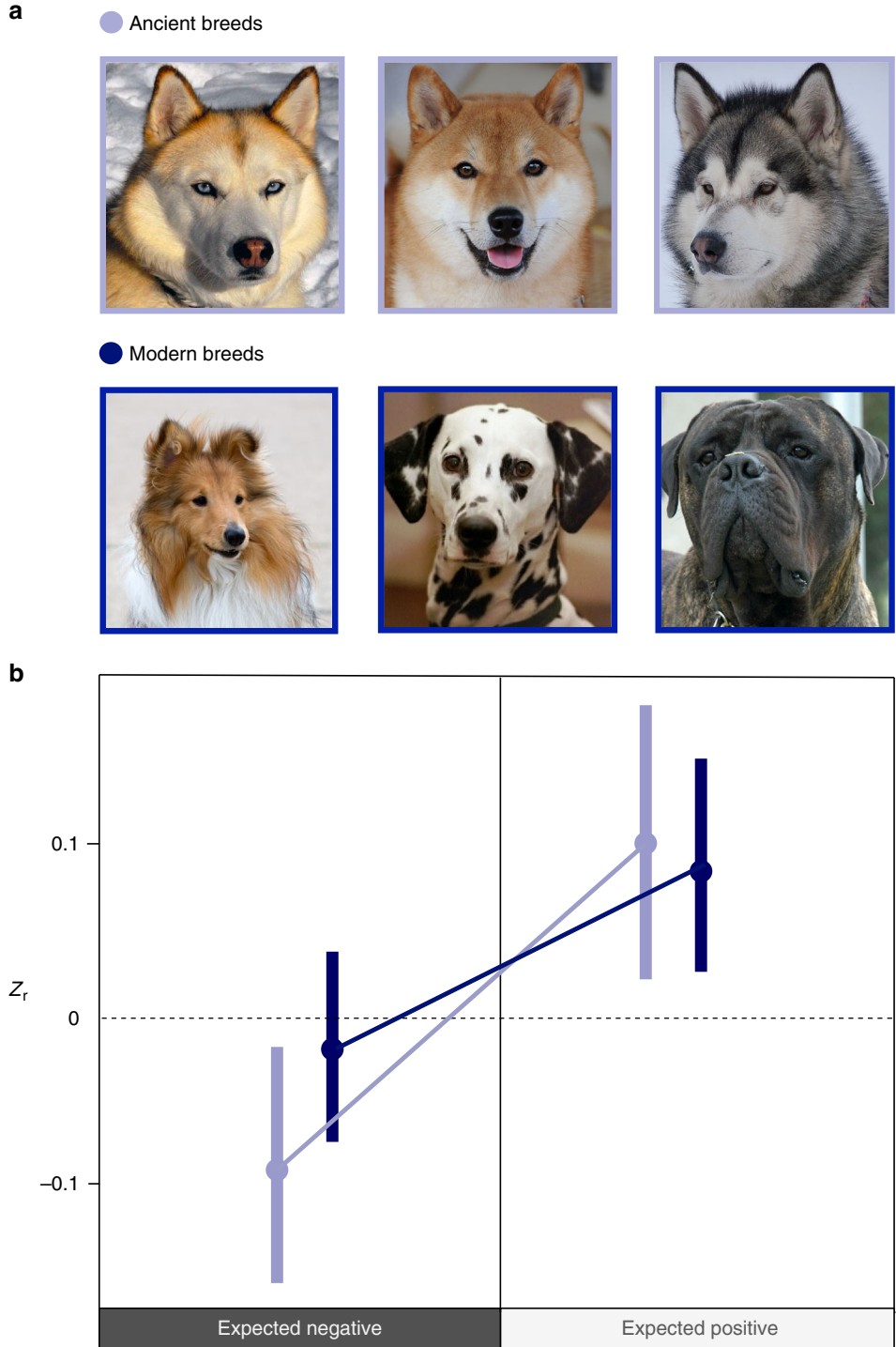

that behavioural correlations predicted by the domestication syndrome hypothesis are decoupled in modern dog breeds.

## Discussion

Using a sampling regime of more than 76,000 dogs, we present a collective examination of the correlations between the key prosocial and reactive behavioural traits hypothesized to change in concert during domestication. We found broad support for the predictions from the domestication syndrome hypothesis. Behavioural correlations hypothesized to be positive during domestication exhibited the expected direction-specific relationships in both modern and ancient breeds, while correlations expected to be negative were only significant in the predicted direction for ancient breeds. Thus, we found that both ancient and modern dog breeds exhibited positive behavioural correlations within prosocial and reactive behaviours, but only ancient breeds expressed exaggerated correlations compared with modern breeds across prosocial and reactive behaviours. These findings suggest that domestication broadly acts on suites of behaviours in concert and indicate that the strength and direction of behavioural correlations in dogs are dependent on the domestication

**Fig. 1** Strength and direction of behavioural correlations between breed categories. **a** Examples of ancient and modern dog breed categories. From top left to right, Siberian Husky, Shiba and Alaskan Malamute as examples of the seven breeds belonging to the ancient breed group, and from bottom left to right, Shetland Sheepdog, and Bullmastiff as examples of the 71 breeds belonging to the modern breed group. **b** Mean $Z_r$ values and 95% confidence intervals for behavioural correlations based on a phylogenetically controlled Bayesian general linear mixed model in ancient modern dog breeds for all expected negative behavioural correlations (between prosocial and reactive behaviours) and expected positive correlations (within prosocial and reactive behaviours). The values for means and 95% confidence intervals are as follows: Mean$_{\text{Expected negative, Ancient}} = -0.088$, 95CI$_{\text{Expected negative, Ancient}} = -0.155$ to $-0.02$; Mean$_{\text{Expected negative, Modern}} = -0.018$, 95CI$_{\text{Expected negative, Modern}} = -0.071$ to $0.03$; Mean$_{\text{Expected positive, Ancient}} = 0.103$, 95CI$_{\text{Expected positive, Ancient}} = 0.015$ to $0.187$; Mean$_{\text{Expected positive, Modern}} = 0.087$, 95CI$_{\text{Expected positive, Modern}} = 0.019$ to $0.158$. Source data for this figure are provided as a Source Data file. Photo credits: Siberian Husky: Original photo taken by Flickr user Sue and Marty. Edited by User: Pharaoh Hound—Flickr. Edited by Pharaoh Hound, CC BY 2.5, https://commons.wikimedia.org/w/index.php?curid=1790050. Shiba: Roberto Vasarri—Own work, Public Domain, https://commons.wikimedia.org/w/index.php?curid=5788123. Alaskan Malamute: waga11, CC BY-SA 3.0, https://commons.wikimedia.org/w/index.php?curid=52965347. Shetland Sheepdog: Karen Arnold—http://www.publicdomainpictures.net/view-image.php?image=38472&picture=dog-portrait, CC0, https://commons.wikimedia.org/w/index.php?curid=26034428. Dalmatian: Black spotted dalmatian—6 y.o. bitch Bobbie—photographed by Nevilley, December 2002, CC BY-SA 3.0, https://commons.wikimedia.org/w/index.php?curid=131558. Bullmastiff: Eran Finkle—Flickr: Dog, Bullmastiff—כלב, בול מאסטיף, CC BY 2.0, https://commons.wikimedia.org/w/index.php?curid=20314170

---

**Table 1 Comparison of direction-specific effect sizes between breed categories**

| Parameter | Estimate | Lower CI | Upper CI | pMCMC |
|---|---|---|---|---|
| (Intercept) | −0.09 | −0.16 | −0.01 | 0.02 |
| Breed category | 0.07 | 0.02 | 0.12 | 0.004 |
| Expected correlation | 0.20 | 0.09 | 0.30 | <0.001 |
| Breed category: expected correlation | −0.09 | −0.15 | −0.02 | 0.006 |
| Phylogeny | 2.24E−04 | 1.98E−08 | 5.75E−04 | – |
| Breed | 1.81E−04 | 2.13E−10 | 4.70E−04 | – |
| Correlation type | 0.007 | 0.003 | 0.013 | – |
| Units | 0.0015 | 0.0012 | 0.0018 | – |

The model output from the Bayesian general linear mixed model on positive vs. negative correlations within and between prosocial and reactive behaviours in ancient and modern breeds. Estimates, 95% confidence intervals (CI) and pMCMC values for the intercept, breed category (ancient or modern), expected correlation (positive or negative), the interaction term between breed category and expected correlation are specified. Random factors are the phylogeny, breed (78 different breeds) and the correlation type (e.g. sociability vs. playfulness)

---

history of the dog breeds being considered (i.e. ancient vs. modern breeds). Therefore, our results imply that that suites of correlated behaviours are relative between breed categories and have been decoupled, both within and between breeds, on an evolutionary scale during dog domestication.

The exaggerated effect sizes between prosocial and reactive behaviours detected in ancient breeds compared with modern breeds suggest that the behavioural architecture may differ in breeds sampled from either early or late stages of dog domestication. As ancient breeds are characterized by genetic admixture with wolves, the wild ancestor of dogs, the reduced effect sizes detected in modern breeds suggest that the process of domestication may decouple some correlated behaviours that were either present in the wild ancestors to dogs, or favoured at earlier stages of dog domestication. Such context-dependent behavioural covariance is commonly observed in comparisons of rural and urban populations, although whether urbanization weakens or strengthens behavioural correlations remains a point of on-going debate (e.g. refs. [46,50,51]). The muted correlation between prosocial and reactive behaviours we observed in modern dog breeds may be caused by recent, stringent selective breeding for specific phenotypes decoupling the expression of some behavioural correlations in modern compared with ancient dog breeds[33]. Our findings suggest that domestication can decouple specific

behavioural correlations and that recent alteration of selection pressures can affect the expression of correlated behaviours in dog breeds.

The exaggeration in effect sizes among ancient breeds were only evident when considering correlations that were a priori expected to be negative based on the domestication syndrome hypothesis. In contrast, the magnitude of effects observed in ancient and modern breeds when considering expected positive correlations within prosocial and reactive behaviours did not differ. Altered selection regimes are often associated with incomplete alterations in behavioural covariance[47,52–54]. For example, in selection lines for either pro-active vs. reactive individuals or bold vs. shy individuals, behavioural covariance was differentially impacted in house mice (*Mus musculus domesticus*[52]) and great tits (*Parus major*[53]), respectively, with changes in some behavioural correlations while others remained unaltered. Similarly, partial decoupling of a behavioural syndrome has been reported in house sparrows (*Passer domesticus*) where some behavioural correlations persist while others break down when comparing urban and rural populations[47]. Differences in the consistency of responses among behavioural correlations suggest that behavioural correlations are relative in part, and may not originate from the same underlying causation[54]. In support of this hypothesis, a recent study assembling and re-sequencing the genomes of conventional, tame and aggressive red fox populations identified multiple genomic regions associated with differential expression of aggressive behaviour across populations[55].

An alternative explanation for our results is that ancient breeds have been subject to a distinct set of selection pressures that have more strongly maintained behavioural correlation associated in the domestication syndrome. For example, if ancient breeds were exposed to similar environments or homogenous selection pressures that differed consistently from modern breeds, then the difference we found between the two breed types could be caused by specific selection to ancient breeds causing behaviours to deviate between breed categories. Such deviations between ancient and modern breeds occur for two of the behaviours we assessed (playfulness and persistent aggression, see Supplementary Table 3). However, these two behaviours did not differ systematically in the direction that generated our results (see Supplementary Figure 1), and are hence unlikely to drive the pattern that we observed. Moreover, the seven ancient breed we examined originate from three geographically distinct origins and vary in contemporary function (i.e. hunting dogs, sledge dogs, companion dogs) and morphology[56]. Direct environmental effects are also unlikely to explain the exaggerated effect sizes in ancient breeds, as dogs belonging to ancient and modern breeds

were sampled in both city and rural environments (see Supplementary Table 2), and the sampling locations were accounted for in our analyses. Thus, it seems unlikely that either environmental-specific selection or direct environmental effects exerted on ancient breeds used in this study caused the shift in behavioural correlations from modern breeds. Instead, the common denominator among ancient breeds is that they, unlike modern breeds, all express a significant detectable admixture with wolf and have an origin 300–400 years before modern breeds[31]. Accordingly, while continuous selection has been exerted on the ancient breeds since their divergence from wolves, they nonetheless represent a proxy for an earlier stage of dog domestication.

Inferring the evolutionary impact of phenotypic correlations relies on the alignment of those correlations with underlying genetic correlations[57]. Generating genetic data in studies on behavioural syndromes can be challenging and given the size of our data set and the time frame (16 years) over which the data were collected, this is a limitation in our study. Given the requirements imposed by breed standards for breed-specific phenotypes, genetic architecture likely varies across breeds and differences in heritability are therefore expected. However, while we cannot infer the underlying genetic correlations in our behavioural traits correlations, phenotypic correlations provide ample information about the direction and magnitude of underlying genetic correlations[57]. Thus, to the extent that our behavioural trait correlations are representative for the underlying genetic correlations, the results in this study provide valuable insight to the evolutionary trajectory of the behavioural correlations in the domestication syndrome.

In sum, our results suggest that behavioural correlations related to domestication can be decoupled, thereby contradicting recent arguments suggesting a single physiological mechanism underlie myriad changes in behaviours, physiology and morphology observed in domesticated animals (i.e. Wilkins et al.[8]). Instead, our results suggest that multiple independent factors act simultaneously to shape the correlated changes observed in the domestication syndrome. To evaluate the extent of the context-specificity of the domestication syndrome, a crucial next step is to quantitatively evaluate if correlations between behaviours, as well as morphology and physiology, are present in other domesticated species while also taking heritability into account. Domesticated animals vary profoundly in their basic ecology and domestication history (e.g. cats[58]), and together with the continuous subjection to strong, species-specific and human-induced directional selection it is likely erroneous to assume that a synonymous domestication syndrome applies to all domesticated animals. However, the increased potential for human–animal interactions in the Anthropocene means that we may have contemporary cases (e.g. dingos, coyotes and even modern wolf populations[59]) where we can evaluate behavioural changes related to domestication in real time.

## Methods

**Study animals.** This study was based on data obtained from the Swedish Kennel Club's database of 76,158 dogs representing 78 different registered dog breeds (range 19–16,243 dogs/breed) tested in the DMA during the years of 1997–2013. Of the 78 dog breeds, seven belong to the ancient breed group ($n = 251$) and 71 belong to the modern breed group ($n = 75,907$) (see vonHoldt et al.[31] for extensive breed grouping based on phylogenetic analyses). The variation in number of breeds within the two groups reflects a biological difference originating from the domestication history of dogs. Specifically, ancient breeds, which originated during earlier stages of dog domestication (> 500 years ago[33]), belongs to a small group of dog breeds with few breeds represented. On the other hand, modern breeds arose from extensive breeding efforts primarily over the past 200 years[33] and represent the vast majority of present day registered dog breeds. Therefore, the sampling difference between the two breed groups in our study was not due to a lack in sampling effort, but because there are far fewer ancient than modern breeds to consider. All dogs belonged to internationally approved breeds and were privately owned pets. It is a strict requirement that only registered, purebred dogs can be enroled in the DMA test, and proof of pedigrees for all dogs tested is therefore mandatory.

The DMA was performed once on both male and female dogs, ranging from 12 months (post sexual maturation) to 4 years, at a range of testing locations across Sweden. In order to consider whether bias existed between breeds in geographical locations of Sweden, we assigned the categories North, South or East as well as rural or city to each dog, based on the owner's geographic location in Sweden (see breakdown of sample sizes based on these categories in Supplementary Table 2). The three geographical categories were based on the three regions of Sweden classified by the European Union and city–rural locations were assigned to testing locations within the city limit of the three large cities in Sweden: Stockholm, Göteborg and Malmö (population of > 300,000[60]). Dogs tested outside the three cities were classified as rural. Behaviours were scored by 316 individual trained observers.

**Behavioural assays.** The DMA is a standardized behavioural assay widely used on dogs in Sweden to quantify and assess a range of behavioural traits in a sequential manner, including sociability, playfulness, fearfulness and aggression[48,49]. The DMA test battery consists of 33 descriptions of behavioural reactions in 10 separate situations (for a full description of the DMA test battery we refer to Svartberg and Forkman[48]). Briefly, in the DMA tests a familiar person, typically the owner, accompanies the dog throughout the tests. A trained test leader walks the owner through all steps of the tests and verifies their correct execution, while a trained observer, not familiar with the dog, assesses the dog's behavioural reaction on an intensity scale from 1 (low intensity) to 5 (high intensity). Previous studies have used all 33 behavioural reactions to uncover possible personality traits in dogs, their stability and broader application[35,48,49,61,62]. For this study we had a clear a priori hypothesis regarding how specific behaviours should change together as a consequence of domestication. We therefore focused on seven specific behavioural traits from the DMA that are directly relevant to the domestication syndrome hypothesis; sociability, playfulness, fearfulness (two measures) and aggression (three measures) (see Supplementary Table 1 for detailed description of the assessment of these behaviours). We focused on within-breed (i.e. among individual) correlations among these seven behavioural traits in our analyses as this is the level where selection is applied during dog domestication in order to target breed-specific behaviours. However, because selection at the within-breed level could influence breed-specific mean values, we also examined variation in correlations among these seven behavioural traits when focusing on behavioural correlations based on breed mean values.

While the individual behavioural scores obtained for this study were based on unique assay events, the behavioural assessments from the DMA are repeatable over time[61] and can therefore be used as proxy measures of behavioural reactions in contexts outside the testing framework (i.e., the average reliability estimates with scores obtained from the DMA and scores obtained when behaviours were assessed in every-day situations were 0.75)[62]. The DMA thus provides an excellent tool for assessing consistent individual behavioural traits within the framework of behavioural syndromes. Furthermore, the DMA offers an excellent alternative to frequently used owner-based questionnaires to estimate behavioural parameters in dogs (e.g. C-BARQ), as the DMA evaluates behavioural reactions measured under standardized testing conditions by certified, non-owner observers. The use of a stranger (i.e. the test leader) provides a comparable basis to engage in social interactions for all dogs during the DMA and removes potential bias that may arise from human–dog familiarity affecting a dog's willingness to engage in social interactions[63]. Observers, test leaders and assistants involved in the practical execution of the DMA test has undergone specific training and are certified by the Swedish Working Dog Association. Specifically, to qualify for the role as a DMA observer scoring the dog's behaviour throughout the test, one must have 125 h of theoretical training and at least 2 years of practical experience with the test. While, the sequential quantification of behaviours in the DMA could bias our results if ancient and modern breeds respond differently to sequential testing, there is no a priori reason to expect such differential responses between breed categories.

**Ethical statement.** According to Swedish legislation, no ethical permission was needed for this study. All dogs were privately owned pets. Dog owners exclusively made the decision to enter their dogs in the DMA test procedure and could retract the dog from the test at any time during the DMA.

**Statistical analyses.** Data on the more than 76,000 dogs used in this study were collected using a standardized approach over a period spanning 16 years. While this type of expansive data set offers unparalleled opportunities to assess broad questions about the domestication of dogs, generating such a large data set necessarily requires behavioural variables to be assessed in standardized way, which loses resolution from finer-scale behavioural measures. Conventional statistical approaches (e.g. ref.[64]) developed for qualitative characterization of data are therefore not well suited for dealing with the ordinal structure of the behavioural data in our analyses. Hence, attempting to apply multivariate approaches to our seven behavioural variables caused considerable convergence problems (including when fit using threshold/ordinal errors) in the resulting, highly parameterized

multivariate models. Further, our large data set introduced variance in the sex, age, location of testing, and identity of the observer into our analyses, which needed to be considered. Indeed, preliminary analyses revealed associations between the behaviours we considered and breed category (ancient vs. modern), sex, age and testing location (Supplementary Table 3). Our explicit interest was in behavioural correlations and how these correlations are influenced by breed category, as predicted by the domestication syndrome hypothesis. Therefore, to evaluate if dogs present behavioural correlations that are consistent with the predictions of the domestication syndrome hypothesis, while accounting for the potential effects of sex, age and testing location, we extracted residual behavioural values from linear mixed effects models performed on the full data set where the aforementioned factors were fitted as predictors and observer identity and breed were treated as random effects. These residual values offer the opportunity to assess behaviour covariance at the within-breed level independently from the effects of sex, age and testing location. This approach assumes equal slopes among breeds across sex, age and testing location, which may not be the case. Therefore, we also evaluated how much additional variation was explained by models allowing for breed-specific random slopes of sex and age (note that for some breeds there was no variation in testing location and therefore we did not examine this factor further). The maximal additional variation explained by these random slope models was low (0.4%), suggesting that our assumption of equal slopes across breeds is acceptable. Thus, analyses were performed on residual behavioural values, which were used to generate a correlation matrix among all behavioural traits using Pearson correlations. This approach produced correlation coefficients (**r**) for all within-breed behavioural correlations for a total of 17 behavioural combinations (Supplementary Data 1, 2 and 3).

After generating correlation coefficients among all pairwise combinations of behaviours, we compared the strength and direction of effects across breeds and breed categories using a Bayesian phylogenetic meta-analysis implemented in the package MCMCglmm[65] in R v.3.4.0 (CRAN), R Studio v.1.0.143. As our aim was to assess broad patterns of behavioural correlations among dog breeds, we had to account for the wide variation in sample size among the 78 breeds examined. To do this, we converted the Pearson correlation coefficients (**r**) from each pairwise behavioural correlation into sample size-weighted Fisher $Z_r$ values and calculated the sampling variance associated with each behavioural correlation and breed[66]. This modelling approach weights effect sizes based on sampling effort and incorporated sampling variance explicitly into our analyses. We used funnel plots to assess sampling bias, but found no such patterns, and the estimated mean and the fitted regression line were identical in all 17 pairwise behavioural correlations (Supplementary Figures 2–4).

An important consideration in any meta-analysis is the independence of the data being examined[67–69]. Among dogs, the history of domestication generates non-independence among breeds in a manner analogous to phylogenetic effects observed among other species. To account for the domestication history of dogs, we used a recent cladogram generated using an identity-by-state distance matrix and a neighbour-joining tree algorithm of 161 dog breeds[32]. This cladogram is the most robust representation of the temporal patterns of dog domestication available currently. In cases where multiple individuals from a breed were sampled, we haphazardly chose one representative individual to generate a reduced breed-specific cladogram for our downstream analyses. Of the 78 breeds considered in our analysis, 12 breeds were not present in this cladogram (American Akita, Beauceron, Smooth Collie, Danish Swedish Farmdog, Dutch Shepherd Dog, Finnish Lapphund, German Pinsher, Hovawart, Lapponian Herder, Medium Poodle, Romagna Water Dog and White Swiss Shepherd Dog). We manually added these breeds to the cladogram by creating polytomies with their closest relative present in the cladogram. To resolve polytomies for analyses, branch lengths were set to arbitrarily small numbers (0.000001). Placement of the 12 missing breeds into the cladogram was achieved by consulting the official breed standards from Fédèration Cynologique Internationale (FCI[56]), which is the largest federation of kennel clubs in the world and provides descriptions of all registered dog breeds, including breed history.

To assess the direction of behavioural correlations and if the strength of effects differs between breed type (ancient vs. modern), we classified the type of pairwise behavioural correlation as either positive (i.e. within prosocial (sociability and playfulness) and reactive behaviours (aggression and fearfulness)), or negative (across prosocial and reactive behavioural categories), based on the direction predicted from the domestication syndrome hypothesis. We then performed a phylogenetically corrected meta-analysis, modelling how $Z_r$ values are influenced by the fixed effects of predicted direction of effects (positive vs. negative), the breed type (ancient vs. modern) and their interaction with the type of behavioural correlations (i.e. the 17 pairwise correlations), the breed and the phylogenetic relationship among species included as random effects, while accounting for sampling variance of $Z_r$. Rather than adding the type of behavioural correlation as a random effect in the model, an alternative approach would be to add both behavioural traits underlying each correlation as random effects to the model to account for the hierarchical nature of the data. However, given the number of behavioural traits ($n = 7$) in our data set, a model that treated the behavioural traits as two separate random effects was overparametrized and unable to disentangle the variance associated with each trait in a model (indicated by high autocorrelation and exaggerated parameter estimates). Therefore, we focused solely on the model described above.

Evolutionary inferences on correlations depend on whether changes in correlations among groups results from changes in covariation or variation[70]. Therefore, we assessed whether changes in correlations were driven by changes in behavioural covariation or variation by extracting covariance and variance values from each of the 17 correlations among behavioural traits within each breed. For the predicted negative correlations, the difference between the ancient and modern breed means of covariances and variances is due to a shift in covariances, not variances: mean covariances differed in magnitude between ancient ($-0.063$) and modern ($-0.010$) breeds, while the square root of the mean variances was similar for ancient (0.729) and modern (0.729) breeds. For the predicted positive correlations, ancient breeds had both slightly lower mean covariance (ancient: 0.061, modern: 0.069) and square root of the mean variance (ancient: 0.838, modern: 0.936) values than modern breeds. Therefore, our main inferential result does not appear to be driven by changes in variance, but is instead consistent with changes in covariance.

To assess if general patterns in the direction and strength of behavioural correlations were driven by behaviour-specific correlations, we next performed a series of exploratory analyses by using separate meta-analytical models for each of the pairwise behavioural correlations. We modelled how $Z_r$ values are influenced by the fixed effect of breed type (ancient vs. modern), while accounting for sampling variance of $Z_r$. We were not able to fit both sampling variance of $Z_r$ and a phylogenetic effect in these models, as these models produced a singularity owing to over-parameterization of the model. As sampling variance had a larger effect on the main effects we chose a conservative approach of only including the sampling variance of $Z_r$ in our models. This analytical choice also allowed us to explicitly incorporate the wide sampling variance among breeds into our analyses.

All MCMCglmm models were run using flat priors for fixed effects and parameter-expanded locally non-informative priors for random effects. Chains were run for 1–3 million iterations, with a burn-in of 10,000–30,000 iterations and sampling every 1000–3000 iterations. Autocorrelations within the models were assessed in intervals between $-0.1$ and 0.1.

Finally, we performed a qualitative analysis to investigate if changes in within-breed behavioural correlations influence correlations in mean behavioural values among-breed. Analyses focusing on mean behavioural values necessarily generate models with lower sample sizes than our main analyses, particularly for ancient breeds. Given the limitations of these models, we focused solely on contrasting the strength of the covariance between behavioural correlations based on breed means between ancient and modern dog breeds. While the reduced sample size in ancient compared with modern dog breeds could be expected to generate more extreme correlation values, one would not expect exaggerated correlation values that are consistently expressed in the a priori directions predicted from the domestication syndrome hypothesis. Therefore, to evaluate if the strength of behavioural correlations based on breed means differed consistently between ancient and modern breeds, we used Exact binomial tests to determine if the behavioural correlations were in the direction predicted by the domestication syndrome hypothesis for ancient and modern breeds and if ancient breeds exhibited exaggerated direction-specific correlation values compared with modern breeds.

**Reporting summary**. Further information on research design is available in the Nature Research Reporting Summary linked to this article.

## Data availability

The data that support the findings of this study are available from the Swedish Kennel Club but restrictions apply to the availability of these data, which were used under license for the current study, and so are not publicly available. Data are, however, available from the authors upon reasonable request and with permission of the Swedish Kennel Club. The source data underlying Fig. 1b, and Supplementary Figures 1–4 are provided as a Source Data file.

## Code availability

The R code used to analyse the data of this study is standard for the packages used and is available from the authors upon reasonable request.

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

## Acknowledgements

We thank the Swedish Kennel Club for the data of the Dog Mentality Assessment. Christopher W. Wheat and Wouter van den Bijl helped with the statistical analyses.

## Author contributions

H.T. conceived the study and C.H.W., J.L.F. and H.T. planned the research approach. The data set was generated for H.T. by the Swedish Kennel Club. J.L.F. and B.R. determined how to analyze the data. C.H.W., J.L.F. and B.R. analysed the data. C.H.W., J.L.F. and B.R. wrote the paper with input from H.T.

## Additional information

**Competing interests:** The authors declare no competing interests.

