## [Peer Review File · Nature Communications]

Reviewers' Comments:

Reviewer #1:

Remarks to the Author:

The Authors investigate if phenotypic correlations (inferred as between-individual correlations) among suites of behavioral traits are different between ancient and modern breeds. Their main hypothesis is that these traits will generally correlate with each other as a consequence of correlated response to selection (domestication syndrome hypothesis). However, they show that this prediction is more likely to hold for ancient breeds, while modern breeds can be characterized by weaker correlations resulting from the decoupled evolution of behaviors during the latter phases of the domestication process.

Although I am not an expert on dog evolution (and on any dog-related topic) and domestication, I assume that the evidence for the independent evolution of originally linked phenotypic traits in relation to domestication is an important novelty component, and I read the paper with great interest. I note however that idea of the "braking down" of behavioral correlations under certain environmental conditions or selection regimes is well known, and research has shown analogue patterns in response to urbanization for example (i.e. behaviors that correlate in natural habitats may no longer show correlation in an urban habitat). Hence, for the wider-scale interpretation of the results, it would be straightforward to point to the general hypothetical/empirical background from the behavioral syndrome literature (e.g. dating back to Andy Sih's great reviews).

I can identify two aspects of the presented data that remained unexplored in the given context. First, I was wondering what the Authors' would predict/detect for the correlation of behavioral traits at the among-breed level (correlations among the breed-specific mean scores). If the link between behavioral traits is decoupled because there is a selection on one behavioral trait but not on others, this would target breed-specific mean values differently which can have consequences for the among-breed correlation as well. Are there differences between modern and ancient breeds in this regard as well?

Although it is evident that some of the within-breed correlations are smaller in the modern than in the ancient breeds, the effect sizes are strongly correlated across the two categories ($r = 0.83$ based on the data presented in Table 1, see a figure attached). If domestication depletes most of the behavioral correlations in modern breeds, I would predict much weaker correlation between the two sets of effect sizes. Without this I cannot exclude that most of the statistical difference found between particular pairs of effects sizes (on Fig 1) is due to the fact that modern breeds correspond to extremely large sample sizes (both in terms of number of individuals and number of breeds) and the errors around these estimates are very small (see further comments along this line below).

I would have further advised along which the Authors could improve their statistics:

1) The main conclusions of the study rely on models in which both breed category (modern vs. ancient) and distance to wolf ("non-independence" of breeds) are simultaneously entered as fixed predictors. These two variables are heavily confounded by definition (i.e. modern breeds are necessarily at higher distance from wolf) that calls for issues about collinearity. When predictors are strongly associated it is impossible to determine their partial explanatory power from a model that includes both. Therefore, it remains difficult to tease apart whether the modern-ancient distinction at a broad scale, or the common ancestry at a finer scale that is responsible for the among-breed variation in effect sizes. I do not think that this is a crucial problem for the Authors' main conclusion, but some clarifications and more careful interpretation might be needed. Furthermore, the proper "phylogenetic" control is not necessarily achieved by the above model using distance to wolf as a fixed predictor, but such effects should be preferably entered to define the residual structure (i.e. phylogenetic least square methods) or as a random effect (phylogenetic mixed model). Based on these phylogenetic approaches, the Authors could test the phylogenetic signal in the data, which could even be done separately for ancient and modern dog breeds allowing the separation of the category/relatedness effects.

2) It is assumed that age- sex- and observer effects are unlikely to pose problems in the analyses. However, given the much smaller sample sizes for ancient breeds than that of modern breeds, biased sampling can easily occur by chance. For example, due to owners' behavior some ancient breeds might be tested at somewhat different ages than some the grand average of the individuals from modern breeds, or they might be evaluated by a smaller subset of observers. This is an empirical question, and instead of relying on pure assumptions for unbiased sampling, it would be more

convincing if the within-breed (and potentially the between-breed) correlations could be obtained from a mixed linear model including all individuals, in which age- and sex- effects are controlled for (as fixed effects), and the hierarchical structure is handled through the random effect structure (breed and observer ID). Additionally, the relatedness matrix of breeds could also be considered in this model.

3) A bootstrap exercise is performed in order to verify that the vast sample size difference between breed categories is unlikely to confound the main results. This is a good strategy, but unfortunately the bootstrap only considers differences in terms of the number of individuals. Ideally, one could also set up a regime for the random sampling, in which only 7 breeds are sampled (randomly) from the modern breeds. In this way, the sample sizes would become more balanced.

4) The Authors use the functions available from the nlme() package to perform meta-analyses. However, this package has not been developed for meta-analyses. The models fitted by the nlme() function particularly assume that the sampling variances are known only up to a proportionality constant. This is not what meta-analytic models do, as they specifically assume that the sampling variances are known. See http://www.metafor-project.org/doku.php/tips:rma_vs_lm_lme_lmer. I suggest using the metafor() or MCMCglmm() packages that can appropriately perform meta-analyses (and can also handle "phylogenetic effects" and account for the hierarchical structure of data)

The paper reads very well, but the supporting material contains several leftovers potentially from earlier versions that make interpretations difficult at some places.

Reviewer #2:

Remarks to the Author:

This is an interesting manuscript dealing with the evolution of behavioral correlations during domestication. Differences in correlation structure among types of breeding (ancient vs. modern) imply the existence of relative rather than absolute constraints on the evolution of behavioral architecture and is of great interest as such, though this aspect is not highlighted in the paper. Multivariate approaches to the data analyses should be applied to understand whether changes in variance rather than trait integration caused these effects. However, as the study does not address genetic variances and covariances directly, there are multiple competing explanations for the results, namely differences in heritability between types, and differences in non-genetic correlations between types causing the reported effects, firm conclusions cannot be drawn.

Introduction

Line 53. Linkage disequilibrium should also be mentioned as a valid explanation. For a good example in the context of among-species differentiation, see Beldade et al. 2002 PNAS.

Line 83-87. I do not understand the rationale for the hypothesized directions: Why should aggressive animals be more fearful when a pleiotropic genetic or hormonal mechanism is implied?

Line 87-88. It would be useful to introduce correlational selection as an explanation for the behavioral correlations described in lines 84-87, rather than just mentioning proximate mechanisms (line 83-84).

Line 87. The hypothesis posited here is too phenomenological for my taste. Please introduce an

adaptive argument explaining why correlational selection pressures affecting trait integration would have changed.

Line 92-93. Perhaps it would be more useful to rephrase such texts to talk about absolute versus relative constraints on the adaptive evolution of behavior, this study representing evidence supporting the latter explanation. See the Beldade et al. 2002 paper mentioned above.

Line 118. Based on each dog assayed once?

Line 129-onwards (Results). Three key issues obstruct a firm interpretation of the results:

1. Changes in correlations are often due to changes in variance. Here, the lower correlations in modern breeds could be caused by a reduction in variance in modern breeds. Authors should therefore also estimate differences in the variance-covariance matrix rather than focusing on correlations alone.
2. Spearman rank correlations should best not be used as non-parametric stats do not connect well with multivariate evolutionary theory tested here.
3. There is no reason to focus on 17 pairs of bivariate correlations in isolation. Evolutionary biologists have developed methods to compare multivariate variance-covariance and correlations matrices from which information can be derived equivalent to what the authors are interested in here (e.g. evolutionary autonomy measures, see Dochtermann & Dingemans 2013 for an application to behavioral syndrome research). I strongly suggest that the overall interpretation and conclusions are based on analyzing differences in aspects of the multivariate correlation matrix

Line 246-onwards (Discussion). The phenotypic correlation (r_P) between two traits equals the genetic correlation (r_G) times the geometric mean heritability of the two traits plus the environmental correlation (r_E) times $(1 - \text{the geometric mean heritability})$ (see Roff 1997 Evolutionary Quantitative Genetics). The authors largely imply that the weaker phenotypic correlations observed in the modern breeds are due to weaker genetic correlation structures, but as the equation shows, there are two alternative explanations:

1. Genetic and environmental correlations often differ (e.g. this is why G and P matrices differ somewhat in Dochtermann's 2011 Evolution paper on behavioral syndromes) even if both don't differ between the types of breed. If so (i.e., $r_G <> r_E$), differences between ancient and modern breeds in geometric mean heritability will cause differences in phenotypic correlations observed here despite the two types not differing in the tightness of the genetic integration of behaviors. A discussion about breed-type specific heritability is thus required. The conclusions should be toned down. In short, this is not sufficient evidence that trait integration has loosened.

Second, genetic correlations indicative of genetic integration of the behaviors might be the same for the two types of breed but environmental correlations (i.e., due to developmental plasticity) might differ between modern and ancient breeds. Those environmental correlations occur when the same environmental factor has pleiotropic effects on multiple traits. It is very likely that ancient and modern breeds experience very different environments (because of how they are used by their owners, and raised by breeders), hence differences in the phenotypic correlations reported here might simply emerge from differences in environmental conditions that the two breed types currently experience. Again, as discussion is required, and conclusions should be toned down, as this mechanism may imply that the conclusions currently drawn may be invalid.

01/03/2018, Signed, Niels Dingemans

Reviewer #3:

Remarks to the Author:

This is a very interesting and original paper.

The authors attempt to use a remarkably large database on dog behavior to test for behavioral correlations predicted by the "domestication syndrome." Specifically, they tested whether sociability was correlated with playfulness and aggression with fearfulness. The results were not supported in modern breeds, but were supported in ancient breeds. They use these findings to argue that the domestication syndrome can become uncoupled so that domestication-related behaviors become independently selected for in modern dogs.

I think this is a very exciting and important paper. I hope it can be published, but I have a few questions I would like to see cleared up before giving my endorsement.

The use of the Working Dog Mentality Assessment (DMA) of the Swedish working dog association (SWDA) provides access to an immense (in this paper now 90,000 dog) behavioral database. For direct behavioral testing of a mammal this is a totally unprecedented scale, and much is gained by using it. But it needs to be kept in mind that, although the test is administered with great seriousness by the SWDA, it was not developed by scientists. Previous studies have, as cited in the paper, provided some evidence of validity for it, but there are nonetheless issues around using a non-scientific instrument that are not removed by the sheer number of animals tested. The DMA is carried out outdoors at sites throughout Sweden by a vast number of trained test-givers (I believe many hundreds, perhaps thousands). Given that the outcome of interest in this paper lies in a comparison of modern breeds to ancient breeds, we need to consider that the modern and ancient breeds may not have been assigned at random to different test-givers and locales. Is it not possible that certain breeds of dogs are more commonly found in the North of Sweden and others in the South, or some in the country and others in the city, and thus may be tested in locations and by test-givers that lead to different criteria, and that this could influence the outcome? This should be addressed. (I thought there was a study by Svartberg which reported that the single largest factor in a factor analysis of scores from the DMA was the identity of the individual carrying out the test, but I cannot now find that study so maybe I'm wrong about that.)

The Intro fails to explain why the particular patterns of correlation (sociability with playfulness and aggression with fearfulness) are expected.

What is the evidence that the dogs are truly of the breeds claimed? It appears to be owner report: If so, what is the basis for believing these reports? Or did the dogs have breed-club papers, or DNA tests?

Reviewers' comments:

Reviewer #1 (Remarks to the Author):

The Authors investigate if phenotypic correlations (inferred as between-individual correlations) among suites of behavioral traits are different between ancient and modern breeds. Their main hypothesis is that these traits will generally correlate with each other as a consequence of correlated response to selection (domestication syndrome hypothesis). However, they show that this prediction is more likely to hold for ancient breeds, while modern breeds can be characterized by weaker correlations resulting from the decoupled evolution of behaviors during the latter phases of the domestication process.

Although I am not an expert on dog evolution (and on any dog-related topic) and domestication, I assume that the evidence for the independent evolution of originally linked phenotypic traits in relation to domestication is an important novelty component, and I read the paper with great interest. I note however that idea of the “braking down” of behavioral correlations under certain environmental conditions or selection regimes is well known, and research has shown analogue patterns in response to urbanization for example (i.e. behaviors that correlate in natural habitats may no longer show correlation in an urban habitat). Hence, for the wider-scale interpretation of the results, it would be straightforward to point to the general hypothetical/empirical background from the behavioral syndrome literature (e.g. dating back to Andy Sih's great reviews).

Response: This is a great point. We have now added information on the impact of urbanization on breaking down behavioural correlations, both to motivate the work in the Introduction (line 93) and to better contextualize our results in the Discussion (lines 210, 211 and 230).

I can identify two aspects of the presented data that remained unexplored in the given context. First, I was wondering what the Authors' would predict/detect for the correlation of behavioral traits at the among-breed level (correlations among the breed-specific mean scores). If the link between behavioral traits is decoupled because there is a selection on one behavioral trait but not on others, this would target breed-specific mean values differently which can have consequences for the among-breed correlation as well. Are there differences between modern and ancient breeds in this regard as well?

Response: This is an interesting idea. Unfortunately, making predictions about how changes in within-breed behavioural correlations might impact among breed correlations is challenging. Break down of behavioural correlations are not directly expected in the domestication syndrome hypothesis, so there is no *a priori* prediction we can draw from this hypothesis. In cases where studies have reported breakdown in behavioural correlations (including the urban/rural comparisons highlighted above), some behaviours are broken down while others are not. As we now highlight in the revised manuscript, there is a mixed set of empirical results how suites of behaviours respond to changing selection pressures (lines 211 and 227). In our study, breed-specific mean values would

only differ between ancient and modern breeds if there were consistent changes in specific behaviours among these breed categories. If changes are variable among breeds (as is the case when comparing behavioural changes between populations, between urban/rural environments etc.) then it is less clear how breed-specific mean values would respond. Nevertheless, the reviewer's comments were intriguing and out of curiosity we did evaluate if there are differences between ancient and modern breeds in correlations between mean behaviours – these analyses revealed that there were differences in breed categories in two of the 16 behavioural correlations we considered (play vs. persistent aggression and fear vs. sudden aggression). These two behaviours are not driving the within-breed responses we detected and interpreting these results is challenging. Therefore, given the murky theoretical framework for comparing within to among breed correlations we have elected to focus our manuscript exclusively on the within-breed correlations, where the level where selection is being applied. However, we are happy to present these analyses in the supporting material at the editor's discretion.

Although it is evident that some of the within-breed correlations are smaller in the modern than in the ancient breeds, the effect sizes are strongly correlated across the two categories ($r = 0.83$ based on the data presented in Table 1, see a figure attached). If domestication depletes most of the behavioral correlations in modern breeds, I would predict much weaker correlation between the two sets of effect sizes. Without this I cannot exclude that most of the statistical difference found between particular pairs of effects sizes (on Fig 1) is due to the fact that modern breeds correspond to extremely large sample sizes (both in terms of number of individuals and number of breeds) and the errors around these estimates are very small (see further comments along this line below).

Response: We respectfully disagree with the reviewer for several reasons. First, since we are assessing the same behaviours in ancient and modern breeds, it is not surprising that the effect sizes are correlated across the breed categories. We would only expect a reduction in the correlation between effect size of ancient and modern breeds if behaviours were broken down to such a degree that previously correlated behaviours were no longer correlated. We do not argue that this is the case in our manuscript.

Second, while there is an overall correlation in effect sizes between ancient and modern breeds in our revised analysis ($r = 0.93$, when using data corrected for age, sex, testing location and observer ID) this correlation is predominantly driven by strong correlation between effect sizes in behaviours predicted to be positively associated ($r = 0.96$), and less strongly correlated in behaviours predicted to be negatively associated ($r = 0.80$). These results are consistent with our expectations and our findings as we do not find a difference in effect sizes between breed categories in *a priori* expected positive correlations but do find differences in *a priori* expected negative correlations. We acknowledge that the way we presented the data in our initial submission did not highlight the dichotomy between predicted positive and negative behavioural correlations. For clarity, our revised manuscript explicitly presents the results based on the different *a priori* predictions from the domestication syndrome hypothesis.

Third, in addition to the correlation coefficient, the slope of the correlations between effect sizes in ancient and modern breeds needs to be considered to understand the relationship between these variables. When looking at all correlations there is an overall positive allometric slope of 1.60 (based on a reduced major axis regression). Among correlations predicted to be negative by the domestication syndrome hypothesis, the slope of the correlation between ancient and modern effect sizes is even more extremely positively allometric (slope = 2.25). In contrast, the magnitude of the allometric slope is reduced among predicted positive behavioural correlations (slope = 1.27). Because we only detected an exaggeration in effect sizes in behavioural correlations that were *a priori* predicted to be negative, the exaggerated slope we report here is consistent with the results presented in the revised manuscript.

Finally, if the effects were driven by differences in sample size between ancient and modern breeds then we would not expect consistent exaggerations in ancient breeds. Certainly, the extensive sampling of modern breeds would reduce sampling error. But at the same time, the comparatively limited sampling of ancient breeds (due to constraints that arise from the domestication history of dogs rather than a lack of sampling effort on our part) should produce larger variance in effect sizes around the 'true' mean. As the 'true' mean is theoretically better approximated by the greater sample size in modern dog breeds then the null expectation is that the ancient breeds sample size should be variable both above and below the 'true' sample size. Therefore, based solely on sampling bias and unequal sample sizes between breed categories, we would not expect consistent differences between breed categories to emerge, and we would not expect those consistent differences to necessarily be in the direction-specific predictions from the domestication syndrome hypothesis.

I would have further advised along which the Authors could improve their statistics:

- 1) The main conclusions of the study rely on models in which both breed category (modern vs. ancient) and distance to wolf ("non-independence" of breeds) are simultaneously entered as fixed predictors. These two variables are heavily confounded by definition (i.e. modern breeds are necessarily at higher distance from wolf) that calls for issues about collinearity. When predictors are strongly associated it is impossible to determine their partial explanatory power from a model that includes both. Therefore, it remains difficult to tease apart whether the modern-ancient distinction at a broad scale, or the common ancestry at a finer scale that is responsible for the among-breed variation in effect sizes. I do not think that this is a crucial problem for the Authors' main conclusion, but some clarifications and more careful interpretation might be needed.

Response: This is a fair point. In the revised manuscript we have removed all analyses that included breed category and distance to wolves in the same model.

Furthermore, the proper “phylogenetic” control is not necessarily achieved by the above model using distance to wolf as a fixed predictor, but such effects should be preferably entered to define the residual structure (i.e. phylogenetic least square methods) or as a random effect (phylogenetic mixed model). Based on these phylogenetic approaches, the Authors could test the phylogenetic signal in the data, which could even be done separately for ancient and modern dog breeds allowing the separation of the category/relatedness effects.

Response: Another good point. We now account for the relationships between dog breeds by including the phylogeny as a random effect in a Bayesian mixed effects model. The reviewer also raises the interesting idea of assessing phylogenetic signal in the data. However, we have not implemented analyses of phylogenetic structure in the revised manuscript as we feel they provide limited scope to address the main questions of our manuscript in a rigorous manner. Specifically, ancient and modern breeds differ in sample size due to the breeding history of dogs. Therefore, since we would necessarily be comparing phylogenetic signal in two groups that differ in sample size it is not clear how any differences or similarities in phylogenetic signal would be interpreted – i.e. are effects/similarities due to the breed category or differences in sample size? Moreover, we are not aware of statistical modelling that addresses the impact of sample size on comparisons of phylogenetic signal among categorical variables. Finally, a recent critique has emphasized that assessing phylogenetic signal when considering low sample sizes and using standard maximum likelihood approaches can generate spurious results, including a biased tendency to support Ornstein-Uhlenbeck models of trait evolution (see Cooper et al. 2016, *Biol J Linn Soc* 118:64-77). While alternative Bayesian methods have been introduced that attempt to account for biases introduced by low sample sizes (see Cooper et al. 2016), the underlying difference in sample size between ancient and modern breeds limits interpretation of these models. Thus, while it would certainly be possible to implement and compare phylogenetic signal among breed categories we think this would not help to resolve the core questions of our work.

2) It is assumed that age- sex- and observer effects are unlikely to pose problems in the analyses. However, given the much smaller sample sizes for ancient breeds than that of modern breeds, biased sampling can easily occur by chance. For example, due to owners’ behavior some ancient breeds might be tested at somewhat different ages than some the grand average of the individuals from modern breeds, or they might be evaluated by a smaller subset of observers. This is an empirical question, and instead of relying on pure assumptions for unbiased sampling, it would be more convincing if the within-breed (and potentially the between-breed) correlations could be obtained from a mixed linear model including all individuals, in which age- and sex- effects are controlled for (as fixed effects), and the hierarchical structure is handled through the random effect structure (breed and observer ID). Additionally, the relatedness matrix of breeds could also be considered in this model.

Response: Another very useful suggestion. This issue was also raised by the other reviewers, with Reviewer 2 suggesting similar analytical approaches to

deal with our data. To deal with this issue, we first ran Bayesian multivariate models that included, age, sex, and testing location as fixed effects and breed, observer ID and phylogeny as random effects, directly in line with the reviewer's suggestion. Unfortunately, these models did not converge – indeed conventional statistical approaches are ill suited for dealing with the ordinal structure of our dataset (see response to Reviewer 2 for more details). Therefore, to account for the effects of age, sex, testing location in our analyses, observer ID, and breed, we extracted residual values from each behavior from linear mixed effects models performed on the entire dataset that included age, sex and testing location as fixed effects and observer ID and breed as random effects. This analytical choice introduced its own issues to consider, as the use of residual values in statistical models can be problematic as they inflate statistical power by reducing degrees of freedom and can lead to biased parameter estimates (see Freckleton 2002 *J Anim Ecol* 71:542-545). However, in this case, we were comfortable using residuals in our analyses because 1) standard statistical approaches designed for multivariable were not able to deal with the structure of our dataset, 2) our sample size (>76,000 dogs) was so large that reductions in degrees of freedom will have exceedingly minor effects on the model fit and 3) the effect size of the correlations among the independent variables were generally weak (ranging from -0.04 to 0.006), and the use of residuals in analyses is only problematic when correlations among predictors is strong. More to the point, analyzing residual values allowed us to test our core prediction of behavioural covariance at the within-breed level independently from the effects of age, sex, and testing location, while also explicitly incorporating potential variance introduced by observer ID and breed. Using these residual behavioural values, we then compared the correlation structure among behaviours between ancient and modern dog breeds while evaluating the *a priori* predictions from the domestication syndrome hypothesis in models that accounted for phylogenetic effects. We now detail these analytical steps in our revised manuscript (lines 133, 137, 385 and 387).

3) A bootstrap exercise is performed in order to verify that the vast sample size difference between breed categories is unlikely to confound the main results. This is a good strategy, but unfortunately the bootstrap only considers differences in terms of the number of individuals. Ideally, one could also set up a regime for the random sampling, in which only 7 breeds are sampled (randomly) from the modern breeds. In this way, the sample sizes would become more balanced.

Response: The bootstrap analysis was done in the original submission because, as the reviewer pointed out in their next comment, the nlme package we used in the original submission did not deal with sampling variance and we were concerned with how variance in sample size might influence our findings. In the revised manuscript, we used meta-analytical models implemented in MCMCglmm that explicitly incorporates sampling variance among breeds. When appropriately implemented, as we now do in the revised manuscript, meta-analyses are designed to account for variation in sample size among data points (i.e. different studies or in this case different breeds) by weighted effect sizes by sample size. Therefore, we elected to not include a bootstrapping set of analyses

in the revised manuscript as our analyses adequately account for variance in sample sizes. Furthermore, one might question the biological relevance of bootstrapping so excessively to match the sample sizes of ancient and modern breeds. In our study we are not limited by sampling effort of ancient breeds, but by the fact that very few ancient breeds exist due to the domestication history of dogs. We now emphasize in the manuscript that the unbalanced sample size between ancient and modern breeds stems from the domestication history of dogs rather than from limited sampling of ancient breeds (line 305 and 310).

4) The Authors use the functions available from the nlme() package to perform meta-analyses. However, this package has not been developed for meta-analyses. The models fitted by the nlme() function particularly assume that the sampling variances are known only up to a proportionality constant. This is not what meta-analytic models do, as they specifically assume that the sampling variances are known. See http://www.metafor-project.org/doku.php/tips:rma_vs_lm_lme_lmer. I suggest using the metafor() or MCMCglmm() packages that can appropriately perform meta-analyses (and can also handle “phylogenetic effects” and account for the hierarchical structure of data)

Response: Good point. All analyses are now performed using the MCMCglmm package.

The paper reads very well, but the supporting material contains several leftovers potentially from earlier versions that make interpretations difficult at some places.

Response: Thank you for catching this. We have revised the Supplementary Materials to deal with this problem.

Reviewer #2 (Remarks to the Author):

This is an interesting manuscript dealing with the evolution of behavioral correlations during domestication. Differences in correlation structure among types of breeding (ancient vs. modern) imply the existence of relative rather than absolute constraints on the evolution of behavioral architecture and is of great interest as such, though this aspects if not highlighted in the paper. Multivariate approaches to the data analyses should be applied to understand whether changes in variance rather than trait integration caused these effects. However, as the study does not address genetic variances and covariances directly, there are multiple competing explanations for the results, namely differences in heritability between types, and differences in non-genetic correlations between types causing the reported effects, firm conclusions cannot be drawn.

Response: There are a number of interesting points raised here and we deal with them specifically in the responses below.

Introduction

Line 53. Linkage disequilibrium should also be mentioned as a valid explanation. For a good example in the context of among-species differentiation, see Beldade et al. 2002 PNAS.

Response: Great point. We have now added this to the introduction (line 50).

Line 83-87. I do not understand the rationale for the hypothesized directions: Why should aggressive animals be more fearful when a pleiotropic genetic or hormonal mechanism is implied?

Response: We agree that we should have unpacked this line of argumentation better. We have now thoroughly addressed this in the last paragraph in the introduction (line 87).

Line 87-88. It would be useful to introduce correlational selection as an explanation for the behavioral correlations described in lines 84-87, rather than just mentioning proximate mechanisms (line 83-84).

Response: We agree that this should be added for clarification and have now done so (line 57).

Line 87. The hypothesis posited here is too phenomenological for my taste. Please introduce an adaptive argument explaining why correlational selection pressures affecting trait integration would have changed.

Response: We now provide an adaptive argument (line 92).

Line 92-93. Perhaps it would be more useful to rephrase such texts to talk about absolute versus relative constraints on the adaptive evolution of behavior, this study representing evidence supporting the latter explanation. See the Beldade et al. 2002 paper mentioned above.

Response: This is an excellent suggestion and we have rephrased the argumentative structure throughout the manuscript accordingly.

Line 118. Based on each dog assayed once?

Response: Yes. All dogs were only tested once (line 319). However, as we point out in the Methods (lines 350 and 352), behaviours assayed in the Dog Mentality Test (DMA) are repeatable over time and predict behavioural reactions in contexts outside of the DMA testing environment.

Line 129-onwards (Results). Three key issues obstruct a firm interpretation of the results:

1. Changes in correlations are often due to changes in variance. Here, the lower correlations in modern breeds could be caused by a reduction in variance in modern breeds. Authors should therefore also estimate differences in the variance-covariance matrix rather than focusing on correlations alone.

Response: Please see our response below about dealing with multivariate models.

2. Spearman rank correlations should best not be used as non-parametric stats do not connect well with multivariate evolutionary theory tested here.

Response: Good point. We now base our analyses on Pearson correlations.

3. There is no reason to focus on 17 pairs of bivariate correlations in isolation. Evolutionary biologists have developed methods to compare multivariate variance-covariance and correlations matrices from which information can be derived equivalent to what the authors are interested in here (e.g. evolutionary autonomy measures, see Dochtermann & Dingemanse 2013 for an application to behavioral syndrome research). I strongly suggest that the overall interpretation and conclusions are based on analyzing differences in aspects of the multivariate correlation matrix

Response: This is an excellent suggestion, and is similar to a point raised by Reviewer 1. In an attempt to apply multivariate methods to assess both variance-covariance and correlation matrices to compare ancient and modern dog breeds we collaborated with an expert in multivariate models (now included as a co-author on the manuscript) and performed multivariate Bayesian analyses in line with the recommendation of the reviewer. When applied to the >76,000 dogs in our dataset, this approach was computationally demanding, with each model taking ~60 weeks of computing time to run (models typically took 3-4 weeks to complete when run in parallel on 20 cores). We attempted relatively simplified multivariate models that only evaluated the behaviours of interest, models that evaluated the behaviours while including phylogenetic effects in the model, and finally more complex models that evaluated the behaviours of interest while also incorporating age, sex, testing location and observer ID (the latter as a random effect). In all cases, models failed to converge and model outputs were autocorrelated. Attempts to overcome these issues by fitting models with threshold/ordinal errors were unsuccessful.

The scale of our dataset offers an unprecedented opportunity to assess broad patterns of behavioural covariance across dog breeds. However, this kind of sampling effort requires behaviours to be collected in a standardized way and necessarily reduces resolution. Conventional statistical approaches, such as those suggested by the Reviewer, are developed for qualitative data and are not suited for dealing with the ordinal structure in our behavioural dataset. Therefore, as much as we agree with and appreciate the Reviewer's comment, it is impossible to implement the suggested models in our analyses, even after substantial effort to do so.

However, the core of the Reviewer's comment remains. We therefore generated residual behavioural values that accounted for differences in age, sex, testing location and observer ID (see response to Reviewer 1), calculated correlation coefficients among behaviours, and performed a single phylogenetically

controlled meta-analysis with sample size corrected effect sizes to directly test the direction-specific predictions from the domestication syndrome hypothesis. This approach represents a hybrid between the optimal analytical path suggested by the reviewers and the constraints of applying multivariate models to our dataset.

Line 246-onwards (Discussion). The phenotypic correlation (r_P) between two traits equals the genetic correlation (r_G) times the geometric mean heritability of the two traits plus the environmental correlation (r_E) times $(1 - \text{the geometric mean heritability})$ (see Roff 1997 Evolutionary Quantitative Genetics). The authors largely imply that the weaker phenotypic correlations observed in the modern breeds are due to weaker genetic correlation structures, but as the equation shows, there are two alternative explanations:

1. Genetic and environmental correlations often differ (e.g. this is why G and P matrices differ somewhat in Dochtermann's 2011 Evolution paper on behavioral syndromes) even if both don't differ between the types of breed. If so (i.e., $r_G \neq r_E$), differences between ancient and modern breeds in geometric mean heritability will cause differences in phenotypic correlations observed here despite the two types not differing in the tightness of the genetic integration of behaviors. A discussion about breed-type specific heritability is thus required. The conclusions should be toned down. In short, this is not sufficient evidence that trait integration has loosened.

Second, genetic correlations indicative of genetic integration of the behaviors might be the same for the two types of breed but environmental correlations (i.e., due to developmental plasticity) might differ between modern and ancient breeds. Those environmental correlations occur when the same environmental factor has pleiotropic effects on multiple traits. It is very likely that ancient and modern breeds experience very different environments (because of how they are used by their owners, and raised by breeders), hence differences in the phenotypic correlations reported here might simply emerge from differences in environmental conditions that the two breed types currently experience. Again, as discussion is required, and conclusions should be toned down, as this mechanism may imply that the conclusions currently drawn may be invalid.

Response: This is a great point. Certainly, genetic and environmental factors (and their interaction) likely influence behavioural correlations in the dog breeds we examined. Given the number of breeds we examined the extent to which heritability differs among breeds remains an open question that we are not able to address in this manuscript. Similarly, it is possible that ancient and modern breeds experienced different environment, but here we are able to evaluate this possibility in part with our dataset by incorporating testing location into our analyses (line 320) and assessing if the proportion of ancient and modern breeds sampled in city and rural environments differed (they did not, see Table S2). But ultimately, as the Reviewer highlights, we are not able to distinguish between these potential explanations with our data and have toned down the Discussion to reflect this.

01/03/2018, Signed, Niels Dingemanse

Reviewer #3 (Remarks to the Author):

This is a very interesting and original paper.

The authors attempt to use a remarkably large database on dog behavior to test for behavioral correlations predicted by the "domestication syndrome." Specifically, they tested whether sociability was correlated with playfulness and aggression with fearfulness. The results were not supported in modern breeds, but were supported in ancient breeds. They use these findings to argue that the domestication syndrome can become uncoupled so that domestication-related behaviors become independently selected for in modern dogs. I think this is a very exciting and important paper. I hope it can be published, but I have a few questions I would like to see cleared up before giving my endorsement.

The use of the Working Dog Mentality Assessment (DMA) of the Swedish working dog association (SWDA) provides access to an immense (in this paper now 90,000 dog) behavioral database. For direct behavioral testing of a mammal this is a totally unprecedented scale, and much is gained by using it. But it needs to be kept in mind that, although the test is administered with great seriousness by the SWDA, it was not developed by scientists. Previous studies have, as cited in the paper, provided some evidence of validity for it, but there are nonetheless issues around using a non-scientific instrument that are not removed by the sheer number of animals tested.

Response: The DMA is an extremely rigorous and standardized behavioural assay that offers a comparable estimate of behaviours across dogs. The assays in the DMA are administered by test leaders who have extensive and accredited experience. In the methods we now elaborated on the rigorous education test observers have to go through in order to assess behavioural expression in the DMA (line 360).

The DMA is carried out outdoors at sites throughout Sweden by a vast number of trained test-givers (I believe many hundreds, perhaps thousands). Given that the outcome of interest in this paper lies in a comparison of modern breeds to ancient breeds, we need to consider that the modern and ancient breeds may not have been assigned at random to different test-givers and locales. Is it not possible that certain breeds of dogs are more commonly found in the North of Sweden and others in the South, or some in the country and others in the city, and thus may be tested in locations and by test-givers that lead to different criteria, and that this could influence the outcome? This should be addressed. (I thought there was a study by Svartberg which reported that the single largest factor in a factor analysis of scores from the DMA was the identity of the individual carrying out the test, but I cannot now find that study so maybe I'm wrong about that.)

Response: Thank you for pointing out these important issues. Similar concerns in were raised by the other Reviewers and we have described our analytical approach to dealing with these issues above. Yes, the DMA is administered by a number of individuals (316 observers in our dataset) and across test locations. Therefore, as stated above, to account for the variance introduced by the expansive number of dogs assayed, we extracted residual behavioural values after incorporated the effects of age, sex, testing location and observer identity. This allowed us to evaluate covariance in behaviours at the within-breed level independently of the factors that the Reviewer raised that may influence our results (lines 133, 137, 385 and 387). Moreover, we now demonstrate that the proportion of ancient and modern dog breeds sampled in city and rural environments did not differ in our dataset (Table S2).

The Intro fails to explain why the particular patterns of correlation (sociability with playfulness and aggression with fearfulness) are expected.

Response: We now addressed this in the last paragraph in the introduction (line 87).

What is the evidence that the dogs are truly of the breeds claimed? It appears to be owner report: If so, what is the basis for believing these reports? Or did the dogs have breed-club papers, or DNA tests?

Response: To enroll in the DMA test dogs are required to be a registered pure breed. Thus, owners must present pedigrees for all dogs prior to testing. We now clarified this in the Methods (line 314).

Reviewers' Comments:

Reviewer #1:

Remarks to the Author:

The MS has undergone a considerable improvement during the revision. The Authors has thoughtfully considered my comments on the previous version and convincingly addressed the majority of them. However, I still have some observations concerning two conceptual aspects and the new analytical design.

1) In their response letter, the Authors argue in detail about the necessity of focusing on the within-breed patterns and the irrelevance of the among-breed correlations. The reasoning is relevant, but it regrettably remains non-transparent for the readers who will see the text of the paper only. From their perspective, based on the information that is currently given in the MS, it might not be that evident that by domestication syndrome, the Authors mean within-breed (among individual) correlations. Confusions may arise because artificial selection targets certain breed-specific traits due to some desired function, and in this context it might be also of interest to test how such selection pressures among-breed patterns. Accordingly, recent selection regimes affecting the mean level of expression in certain traits but not in others will break among-breed correlations but not necessarily the within-breed correlations. More motivation should be given to the general readers about the need of focusing on the within-breed patterns when testing the working hypothesis of this paper.

2) From line 232 the Authors conclude that "in the context of the domestication syndrome hypothesis, our results suggest that different genes and/or hormonal pathways are causing the correlations between these behaviours and that the behavioural correlations associated with domestication may, to some extent, not share a simple common mechanistic origin". I think that given that the Authors do not investigate mechanisms and cannot separate genetic effects, this remains a speculation. The fact that some correlations have been weakened does not necessarily mean that the corresponding traits were originally governed by different causative pathways. They may also have had the same mechanistic origin, but selection might have broken up this shared hormonal/genetic link during the evolutionary history.

3) I welcome that the Authors adopted the phylogenetic-meta-analytic framework available in MCMCglmm, but some details of the analyses need to be revised:

a) First, using the entire data the Authors calculate the residuals of the behavioural scores and then estimate the pairwise correlation between these residuals separately for each breed. This can be misleading for two reasons. First, combining the entire data for the first analysis will lead to models in which within- and between-breed effects are mixed. For example, if age differently affects a behavioural trait in one breed than in another, it is not correct to rely on the same general equation when calculating the residuals. Second, it requires a "stat on stat" approach, while a more parsimonious solution would be available. My suggestion is to create a single model for each bred and pair of traits, in which the original scores are used (one as a predictor and one as a response) and the confounders are included at once. From this model, the relevant slope and its standard error can be used to estimate the desired effect size (r)

b) I am not sure how the Authors dealt with autocorrelation. The behavioural traits are assessed in a strict sequence after each other (if I understand correctly). Thus it might be that behaviour in one test can determine the behaviour in the next test situation (e.g. by the level of stress that individuals experience). Thus, correlations between these traits might simply reflect these carryover effects over the test situations. Accordingly, modern and ancient breeds might differ in this autocorrelation component without any consequence for the domestication syndrome.

c) I do not agree with the construction of the main model. From table 1 and from the description, I understand that breed-specific correlations (16 for each breed considering all pairwise combinations) were the response variable, and breed category, expected correlation (i.e. within or between prosocial and reactive categories) and their interaction were the fixed predictors, while breed, phylogeny and correlation type (e.g sociability v. playfulness) were the random effects. I think, this model does not appropriately cover the hierarchical structure of the data. Breeds are nested within the two breed categories, while breed categories cover deep level phylogenetic effects. Hence, ideally these hierarchies should be covered by the random part of the model. The main interest is to estimate the amount of variance that is accounted by breed categories on top of the breed and phylogenetic effects. Furthermore, the pseudo-replication induced by the fact that same traits are used to obtain different correlations should be accounted by including both traits in the correlation as random factors (e.g. sociability for trait1 and playfulness for trait2) instead of using correlation type (e.g sociability-playfulness). From this model, it might be difficult (technically, it is possible) to obtain grouped random effect estimates separately for the two "within or between prosocial and reactive" categories (that is equivalent for the interaction between Breed category:Expected correlation), but this could be achieved by creating two models relying on either the within- or the between-category correlations.

I must note in addition that Table 1 is confusing. I am not sure what "Dog" means in this model, since the unit of this analysis is the breed-specific correlation, so the lowest hierarchical level in the underlying data is the breed level and not individual level. Furthermore, it is supposed to be a result of a phylogenetically inferred model, while random effect for phylogeny is not given (Breed ID is different than phylogeny effect).

d) I am confused by the statement from line 441 "We were not able to fit both sampling variance of Zr and a phylogenetic effect in these models due to the fact that each model only had one observation per breed." I do not think it is true. Sampling variance of Zr should be $1/N-3$, where N is the number of individuals used to estimate Zr. So I am not sure how Vz is calculated and accounted for in the models.

Reviewer #2:

Remarks to the Author:

Authors have submitted an extensively revised manuscript based upon reviewer comments. By doing so they have addressed most but not all key reviewer comments.

1. A key comment was that authors should use multivariate mixed-effects models that control for various biasing effects to estimate the within-breed variances and covariances (hence correlations). Authors clarify that this task was not possible computationally, which is surprising. Nevertheless, authors have now used univariate approaches to estimate values controlling for bias which they then correlated across traits, which seems an appropriate pragmatic solution. My only comment here is that in the results this shouldn't be called "residual correlation" but rather within-breed among-individual correlation (calculated off residuals or something of that kind).

2. A second key comment was that changes in correlation structures can be caused by changes in variances not covariances: $r = \text{covAB}/\sqrt{\text{varA}*\text{varB}}$. The request was to therefore additionally estimate changes in variances and covariances (see e.g. Dingemanse et al. 2007 J Anim Ecol for an early paper appreciating this problem). The authors have not taken up this task. Consequently, based on the analyses presented we cannot ascertain that the main conclusions stand (tighter covariance structure in certain types of breed). Possibly, covariances were unchanged between types of brood

because variances were breed-specific, leading to changed correlations. In my view these additional analyses are required to forcefully interpret the data.

3. A third key argument is that authors should use a multivariate approach to multivariate data rather than presenting sets of pairwise correlations. It is not clear to me whether the authors have taken this comment on board. If they must present sets of pairwise correlations, then it would be appropriate to control for repeated testing of the same data.

4. The "a priori" hypothesis of certain correlations being positive and others being negative is insufficiently detailed. Authors provide insufficient arguments making their case, leading to the feeling that the hypothesis was a postiori rather than a priori formulated.

Reviewer #3:

Remarks to the Author:

All my concerns have been clearly and succinctly addressed. I think the paper should be published in the present version.

Reviewers' comments:

Reviewer #1 (Remarks to the Author):

The MS has undergone a considerable improvement during the revision. The Authors has thoughtfully considered my comments on the previous version and convincingly addressed the majority of them. However, I still have some observations concerning two conceptual aspects and the new analytical design.

1) In their response letter, the Authors argue in detail about the necessity of focusing on the within-breed patterns and the irrelevance of the among-breed correlations. The reasoning is relevant, but it regrettably remains non-transparent for the readers who will see the text of the paper only. From their perspective, based on the information that is currently given in the MS, it might not be that evident that by domestication syndrome, the Authors mean within-breed (among individual) correlations. Confusions may arise because artificial selection targets certain breed-specific traits due to some desired function, and in this context it might be also of interest to test how such selection pressures among-breed patterns. Accordingly, recent selection regimes affecting the mean level of expression in certain traits but not in others will break among-breed correlations but not necessarily the within-breed correlations. More motivation should be given to the general readers about the need of focusing on the within-breed patterns when testing the working hypothesis of this paper.

Response: We see the reviewer's point. We have now added throughout the manuscript that our main point of focus is on within-breed patterns of covariance as this is where selection has been applied during the domestication process of dogs (lines 56-58, 156, 171-174, 195-200, 365-367). However, we gave the reviewer's comments further thought in light of their comments here and in the first round of review. It is now clear to us that the reviewer's point is spot on and that the focus on selection changing breed-specific traits should also influence among-breed patterns of covariance. Therefore, we have included an additional set of analyses that focus on among-breed mean correlation values between behaviours and we contrast these correlations between ancient and modern breeds (lines 195-200, Table S8). The

results from these mean correlation value analyses support our main results from the within-breed analyses. Namely, as the reviewer suspected, break downs in behavioural covariance in modern dog breeds at the within-breed level of analysis are mirrored by break downs in behavioural covariance at the among-breed level. However, while selection during the last century has led to a significantly large number of modern breeds, ancient breeds belong to a small group with only few breeds represented. This low sample size (for among breed correlations) limits the application of multivariate models to our dataset (as we detailed in the last round of review), and we therefore present only a qualitative analysis to compare patterns of covariance among-breeds in ancient and modern dog breeds (detailed in lines 508-522). And in light of Reviewer 2's comments about applying multivariate approaches, we have refrained from over-analyzing this data and have instead synthesized the broad patterns using binomial tests. We view this analysis as interesting, but given the limitations of this approach we do not rely on it as our main source of inference. Nevertheless, we thank the reviewer for this interesting idea and by addressing their query we feel our results are stronger now.

2) From line 232 the Authors conclude that “in the context of the domestication syndrome hypothesis, our results suggest that different genes and/or hormonal pathways are causing the correlations between these behaviours and that the behavioural correlations associated with domestication may, to some extent, not share a simple common mechanistic origin”. I think that given that the Authors do not investigate mechanisms and cannot separate genetic effects, this remains a speculation. The fact that some correlations have been weakened does not necessarily mean that the corresponding traits were originally governed by different causative pathways. They may also have had the same mechanistic origin, but selection might have broken up this shared hormonal/genetic link during the evolutionary history.

Response: This is a good point. We have now removed this sentence from the manuscript.

3) I welcome that the Authors adopted the phylogenetic-meta-analytic framework available in MCMCglmm, but some details of the analyses need to be revised:

a) First, using the entire data the Authors calculate the residuals of the behavioural scores and then estimate the pairwise correlation between these residuals separately for each breed. This can be misleading for two reasons. First, combining the entire data for the first analysis will lead to models in which within- and between-breed effects are mixed. For example, if age differently affects a behavioural trait in one breed than in another, it is not correct to rely on the same general equation when calculating the residuals. Second, it requires a “stat on stat” approach, while a more parsimonious solution would be available. My suggestion is to create a single model for each breed and pair of traits, in which the original scores are used (one as a predictor and one as a response) and the confounders are included at once. From this model, the relevant slope and its standard error can be used to estimate the desired effect size (r)

Response: We respectfully disagree with the reviewer's suggestion for a variety of reasons. We acknowledge that there are many different options available that can be

applied to the kind of modeling approaches we use and we thought long and hard about these options before we implemented the approach we used. We even considered the very approach that the reviewer is suggesting but decided against it. As we argue below, we feel that the modelling approach that we used more adequately deals with the strengths and limitations of the dataset than the approach suggested by the reviewer and further highlight that our approach was viewed as a ‘pragmatic’ solution to deal with the complexities of the dataset by Reviewer 2.

We feel that the modelling approach suggested by the reviewer is not ideal for our dataset because:

1. The model suggested by the reviewer assumes that there is no error in the predictor (which is one of the behaviours in the model suggested by the reviewer). Yet for the behaviours in our dataset, we have an unknown level of error in the predictor, which is common in behaviours. Moreover, these behaviours are measured on a nominal scale (which is necessary when collecting such a large dataset), which would likely increase error as we do not have absolute measurements.
2. If we used the approach suggested by the reviewer, then we cannot include the potential confounding variable for any breed that lacks variance in one of the predictors (in actuality this only applies to the effect of testing location for a subset of breeds). Importantly, not incorporating all of the potentially confounding variables in the model from some breeds would directly contravene the very helpful recommendations from the reviewer in the last round of revisions.
3. The idea that there may be different slopes (i.e. age differentially affecting behavioural traits in different breeds) in the effects is an interesting one. The standard way of modelling different slopes related to a random effect would be to use a random slope model that accounts for all possible interactions with breed. Such a model would be highly parameterized, and these kinds of models are usually not used/recommended for inferential purposes. In our dataset, if the null hypothesis that age has a similar effect on traits in each breed is true, then the combination of potential variation in mean age across breeds with the error of the measurement for each behaviour could potentially yield variation in slopes that are not based on an underlying biological signal, but instead derived from variance in the covariate. Therefore, to avoid creating an over parameterized model, while also addressing the reviewer’s concerns, we assessed the variance explained by random slopes of age and sex on each behaviour. In order to assess our assumption of equal sex and age effects across breeds, we quantified the variance explained by trait specific univariate models including either, 1) just a random effect of breed 2) allowing for breed-specific random slopes of age on the response, and 3) allowing for breed-specific random slopes of sex on the response. We found that the addition of random slopes only increased the fit of the model marginally, with the maximal improvement of fit for our 7 traits was 0.4%. We are therefore confident in that the assumption of equal slopes did not distort the estimates we present.

While we appreciate the issue raised by the reviewer, we are concerned that allowing for different breed-specific slopes may lead to differences in slopes that are generated

through measurement error in the covariate in combination with different sample sizes among breeds. We therefore feel that assuming constant slopes among breeds is a more appropriate approach. The marginal improvement in fit for random slope models that we demonstrate above suggest that random slopes are not driving the results presented in our manuscript. Therefore, we have elected to retain our original modelling approach in the revised manuscript. However, we now add additional explanatory text to the Methods section that highlights the issue raised by the reviewer and explains why we used the modelling approach we employed (lines 416-422).

b) I am not sure how the Authors dealt with autocorrelation. The behavioural traits are assessed in a strict sequence after each other (if I understand correctly). Thus it might be that behaviour in one test can determine the behaviour in the next test situation (e.g. by the level of stress that individuals experience). Thus, correlations between these traits might simply reflect these carryover effects over the test situations. Accordingly, modern and ancient breeds might differ in this autocorrelation component without any consequence for the domestication syndrome.

Response: Yes, we agree with the reviewer that the sequential nature of the test is potentially problematic. However, performing these tests in a sequential manner is a necessary by-product of creating a standardized test that can be widely applied at a large scale (in our dataset more than 76,000 dogs were assayed). The validity of the behavioural test scores generated from the sequential tests in the DMA has previously been assessed against behavioural measures taken in everyday situations with dogs as they interact with their owners. Specifically, Svartberg (2005) compared DMA scores with independent behavioural assessments outside of the DMA testing situation for nearly 700 dogs from 16 breeds and found high correlations between the behavioural scores derived in the DMA and those from every-day life situations (average reliability estimates = 0.75). This relevant information is now highlighted in the Methods section (lines 375-376) and suggests that any potential carryover effects are not swamping the biologically meaningful measures of behavioural variance that the DMA produces.

Further, for sequential effects to influence our results they would have to differentially influence ancient and modern breeds. We feel this unlikely to be an explanation – we have good reason to think that changes in selection due to domestication influence behaviours, while we have no reason to expect order effects in the DMA to differ between ancient and modern breeds. Hence, we have no *a priori* reason to expect that the order effect of the behavioural tests in the DMA should differ between ancient and modern breeds. However, this is an impossible issue to disentangle given our dataset. Therefore, we include a caveat in the manuscript to specify that our analysis assumes that ancient and modern breeds respond similarly to the sequential nature of the behavioural tests (lines 388-391).

c) I do not agree with the construction of the main model. From table 1 and from the description, I understand that breed-specific correlations (16 for each breed considering all pairwise combinations) were the response variable, and breed category, expected correlation (i.e. within or between prosocial and reactive categories) and their interaction were the fixed predictors, while breed, phylogeny and correlation type (e.g. sociability v. playfulness) were the random effects. I think, this

model does not appropriately cover the hierarchical structure of the data. Breeds are nested within the two breed categories, while breed categories cover deep level phylogenetic effects. Hence, ideally these hierarchies should be covered by the random part of the model. The main interest is to estimate the amount of variance that is accounted by breed categories on top of the breed and phylogenetic effects. Furthermore, the pseudo-replication induced by the fact that same traits are used to obtain different correlations should be accounted by including both traits in the correlation as random factors (e.g. sociability for trait1 and playfulness for trait2) instead of using correlation type (e.g sociability-playfulness). From this model, it might be difficult (technically, it is possible) to obtain grouped random effect estimates separately for the two “within or between prosocial and reactive” categories (that is equivalent for the interaction between Breed category:Expected correlation), but this could be achieved by creating two models relying on either the within- or the between-category correlations.

Response: We thank the reviewer for their comment and generally agree with their concern. In fact, we initially constructed exactly such a model but did not present this in the revised manuscript, which we describe in more detail below. Clearly, we should have been more transparent in how we were dealing with the complexities of this model. We have now added an explanation to the Methods section which describes why we used the model that we did in our analyses (Lines 469-476).

The suggested model is problematic given our data structure. There are just seven behavioural traits, yielding in total 17 unique correlations. Adding random effects in pairs to a model using this low number of random effect levels generated unacceptably high autocorrelation values in the posterior chain. This is not surprising given the structure of the data. Moreover, increasing the chain length and thinning to a very large extent (e.g. thinning of 5000, which far exceeds most standard model structures) did not solve the issues with autocorrelation present in the model output. In a model containing both traits (i.e. trait 1 and trait 2) and breed ID, there are very few observations underlying each random effect estimate. The fact that the suggested model is clearly unstable (as evident from the high autocorrelation) means it is not desirable to present such a model in our manuscript. Despite these issues, when we ran the model suggested by the reviewer we found it to be congruent with our main results in terms of being consistent with the significance and direction of our results. However, the suggested model estimates means for ancient/modern breeds in the expected positive and negative direction that were nearly twice the magnitude of the observed means from the raw data. In contrast, the model that we present in the manuscript had acceptable levels of autocorrelation and means that were congruent with those of the raw data. We now acknowledge that the alternative modelling approach suggested by the reviewer is a better way to deal with dependencies in our dataset but also highlight the limitations of applying this model due to the structure of our data (lines 469-476). We have therefore kept the model that we presented in our last submission in the revised manuscript.

One option would be for us to add the model output from the model suggested by the reviewer to the supporting material, while retaining the model we favour in the main text. However, we are reluctant to include the model suggested by the reviewer in the supporting material as the instability of this model makes us wary about drawing inferences from the output. Nevertheless, if the reviewer and editor feel strongly that

this should be included in the manuscript, we are happy to add this information to the supporting material.

I must note in addition that Table 1 is confusing. I am not sure what “Dog” means in this model, since the unit of this analysis is the breed-specific correlation, so the lowest hierarchical level in the underlying data is the breed level and not individual level. Furthermore, it is supposed to be a result of a phylogenetically inferred model, while random effect for phylogeny is not given (Breed ID is different than phylogeny effect).

Response: Thanks for catching this. This was a typo. Where we wrote ‘dog’ it should have been the random effect for phylogeny. We have now corrected the table.

d) I am confused by the statement from line 441 “We were not able to fit both sampling variance of Zr and a phylogenetic effect in these models due to the fact that each model only had one observation per breed.” I do not think it is true. Sampling variance of Zr should be $1/N-3$, where N is the number of individuals used to estimate Zr. So I am not sure how Vz is calculated and accounted for in the models.

Response: Thanks for pointing this out. Our initial description was not accurate and we have now clarified this in the Methods (lines 496-498). Sampling variance of Zr was calculated in the way described by the reviewer. The problem was that the model produced a singularity, due to overparameterization of the model, which prevented us from fitting both sampling variance and phylogenetic effects.

Reviewer #2 (Remarks to the Author):

Authors have submitted an extensively revised manuscript based upon reviewer comments. By doing so they have addressed most but not all key reviewer comments.

1. A key comment was that authors should use multivariate mixed-effects models that control for various biasing effects to estimate the within-breed variances and covariances (hence correlations). Authors clarify that this task was not possible computationally, which is surprising. Nevertheless, authors have now used univariate approaches to estimate values controlling for bias which they then correlated across traits, which seems an appropriate pragmatic solution. My only comment here is that in the results this shouldn't be called "residual correlation" but rather within-breed among-individual correlation (calculated off residuals or something of that kind).

Response: Thanks for the positive comments and suggestion for how to clarify the terminology. We now make sure that the term ‘residual correlation’ is not used in the manuscript and clarified that we are referring to within-breed, among-individual correlations calculated from residual values (Lines 149-150).

2. A second key comment was that changes in correlation structures can be caused by changes in variances not covariances: $r = \text{covAB}/\sqrt{\text{varA}*\text{varB}}$. The request was to therefore additionally estimate changes in variances and covariances (see e.g. Dingemanse et al. 2007 J Anim Ecol for an early paper appreciating this problem). The authors have not taken up this task. Consequently, based on the analyses presented we cannot ascertain that the main conclusions stand (tighter covariance

structure in certain types of breed). Possibly, covariances were unchanged between types of brood because variances were breed-specific, leading to changed correlations. In my view these additional analyses are required to forcefully interpret the data.

Response: Thanks, this is a great point and we apologize for not adequately dealing with this important issue in our last submission. To explore the relative importance of changes in variances/covariances, we extracted the covariance and variances that underlie each of the 16 correlations that we examined for each breed. We subsequently took the mean of the breed specific variances and covariances for each combination of predicted direction and ancient vs. modern breed category. We found that the difference between ancient and modern breeds for the predicted negative correlations is due to a shift in covariances, as the covariances differ greatly in magnitude between ancient and modern (-0.063 vs -0.010 in ancient vs modern breeds, respectively). In contrast, the variances for predicted negative correlations remain basically the same for ancient and modern breeds (sqrt variance product, ancient: 0.7290, modern: 0.7292). For the predicted positive correlations, which did not differ between breed categories, ancient breeds have both slightly lower covariance (ancient: 0.061, modern: 0.069) and variance (ancient: 0.838, modern: 0.936). Therefore, our main inferential result does not appear to be driven by changes in variance, but rather is caused by changes in covariance. We now address this concern in the Methods (Lines 478-490).

3. A third key argument is that authors should use a multivariate approach to multivariate data rather than presenting sets of pairwise correlations. It is not clear to me whether the authors have taken this comment on board. If they must present sets of pairwise correlations, then it would be appropriate to control for repeated testing of the same data.

Response: We agree that it is important to use a multivariate approach and should have been more explicit in how we treated the pairwise correlations from an inferential basis in our revised manuscript. These bivariate analyses were meant to be illustrative to show if any potential differences between ancient and modern breeds were driven by either by general changes in effect sizes between these groups or by behaviour-specific correlations of large effect. Therefore, to improve the presentation of our results and resolve this issue, we have now removed Figure 2 from the main manuscript and moved this to the supplemental material (now Figure S1), we have removed the statistical comparisons between ancient and modern breeds from Table S7 and not referred to direct statistical comparisons in the main text.

4. The "a priori" hypothesis of certain correlations being positive and others being negative is insufficiently detailed. Authors provide insufficient arguments making their case, leading to the feeling that the hypothesis was a postiori rather than a priori formulated.

Response: We have now extensively revised the final paragraph in the introduction to elaborate in detail on why we have *a priori* expectations about the direction of relationships between prosocial and reactive behaviours. Broadly speaking, there are two lines of argumentation here. First, there is the observation that domesticated animals commonly are more social/playful and less aggressive/fearful, which suggests

correlative responses in these traits following domestication. Second, in domesticated species that have been well studied, selection for tameness behaviour in selection experiments leads to increases in social and play-like interactions with humans as well as alterations of endocrinology (i.e. reduced glucocorticoids) and brain neurochemistry that are associated with reduced aggression and fear response. Therefore, there is an *a priori* expectation based on the literature associated with the domestication syndrome hypothesis that prosocial behaviours should be positively co-expressed, reactive behaviours should be positively co-expressed, and prosocial and reactive behaviours should be negatively co-expressed.

Reviewer #3 (Remarks to the Author):

All my concerns have been clearly and succinctly addressed. I think the paper should be published in the present version.

Response: We thank the reviewer for their positive comment.

Reviewers' Comments:

Reviewer #1:

Remarks to the Author:

My concerns have been adequately addressed. Thank you for the stimulating discussion.

Reviewer #2:

Remarks to the Author:

The authors have done an appropriate job revising their manuscript in response to my final comments.
I have no further comments.